# An economic evaluation of *Wolbachia* deployments for dengue control in Vietnam

**Hugo C. Turner**[1]*, **Duong Le Quyen**[2], **Reynold Dias**[3], **Phan Thi Huong**[4], **Cameron P. Simmons**[3], **Katherine L. Anders**[3]

**1** MRC Centre for Global Infectious Disease Analysis, School of Public Health, Imperial College London, London, United Kingdom, **2** World Mosquito Program, Ho Chi Minh City, Vietnam, **3** World Mosquito Program, Monash University, Clayton, Australia, **4** Department of Preventive Medicine, Ministry of Health, Hanoi, Vietnam

* hugo.turner@imperial.ac.uk

**Data Availability Statement:** All relevant data are within the manuscript and its Supporting Information files.

## Abstract

### Introduction

Dengue is a major public health challenge and a growing problem due to climate change. The release of *Aedes aegypti* mosquitoes infected with the intracellular bacterium *Wolbachia* is a novel form of vector control against dengue. However, there remains a need to evaluate the benefits of such an intervention at a large scale. In this paper, we evaluate the potential economic impact and cost-effectiveness of scaled *Wolbachia* deployments as a form of dengue control in Vietnam–targeted at the highest burden urban areas.

### Methods

Ten settings within Vietnam were identified as priority locations for potential future *Wolbachia* deployments (using a population replacement strategy). The effectiveness of *Wolbachia* deployments in reducing the incidence of symptomatic dengue cases was assumed to be 75%. We assumed that the intervention would maintain this effectiveness for at least 20 years (but tested this assumption in the sensitivity analysis). A cost-utility analysis and cost-benefit analysis were conducted.

### Results

From the health sector perspective, the *Wolbachia* intervention was projected to cost US $420 per disability-adjusted life year (DALY) averted. From the societal perspective, the overall cost-effectiveness ratio was negative, i.e. the economic benefits outweighed the costs. These results are contingent on the long-term effectiveness of *Wolbachia* releases being sustained for 20 years. However, the intervention was still classed as cost-effective across the majority of the settings when assuming only 10 years of benefits

### Conclusion

Overall, we found that targeting high burden cities with *Wolbachia* deployments would be a cost-effective intervention in Vietnam and generate notable broader benefits besides health gains.

**Funding:** HCT has received funding from the World Mosquito Program (WMP) to conduct this analysis. HCT acknowledges funding from the MRC Centre for Global Infectious Disease Analysis (reference MR/R015600/1), jointly funded by the UK Medical Research Council (MRC) and the UK Foreign, Commonwealth & Development Office (FCDO), under the MRC/FCDO Concordat agreement and is also part of the EDCTP2 programme supported by the European Union. KLA, CPS, RD and DLQ acknowledge funding from the Wellcome Trust for this work (224459/Z/21/Z). The funders had no role in study design, data collection and analysis, decision to publish, or preparation of the manuscript.

**Competing interests:** I have read the journal's policy and the authors of this manuscript have the following competing interests: HCT received funding from the World Mosquito Program to conduct this analysis. DLQ, RD, CPS and KLA are employees of the World Mosquito Program. PTH has declared that no competing interests exist.

## Author summary

Dengue is a major public health challenge and a growing problem due to climate change. The release of *Aedes aegypti* mosquitoes infected with the intracellular bacterium *Wolbachia* is a novel form of vector control against dengue. However, there remains a need to evaluate the health and economic benefits of such an intervention at a large scale, as well as its value for money. In this paper, we evaluate the potential economic impact and cost-effectiveness of scaled *Wolbachia* deployments as a form of dengue control in Vietnam–targeted at the highest burden urban areas. Ten settings within Vietnam were identified as priority locations for potential future *Wolbachia* deployments (using a population replacement strategy). We assumed that the effectiveness of *Wolbachia* deployments in reducing the incidence of symptomatic dengue cases would be 75%. We found that targeting high burden cities with *Wolbachia* deployments would be a cost-effective intervention in Vietnam and generate notable broader benefits besides health gains. Overall, this work highlights the value of investment in the scaled implementation of *Wolbachia* deployments as an effective and cost-effective tool for dengue control in Vietnam, and more generally for addressing the global challenge of dengue control.

## Introduction

Dengue is a mosquito-borne, acute febrile illness that is a major public health problem in the tropics and subtropics worldwide. Concerningly, its geographical range and incidence are predicted to increase further due to climate change and urbanization [1].

The release of *Aedes aegypti* mosquitoes infected with the intracellular bacterium *Wolbachia* is a novel strategy for the control of dengue and other Aedes-borne diseases [2]. Mosquitoes infected with *Wolbachia* (*w*Mel strain) 1) are less likely to transmit dengue, chikungunya, Zika, and yellow fever viruses [2] and 2) can suppress or replace the natural mosquito population due to fatal cytoplasmic incompatibility among *Wolbachia* wild-type mating pairs [2]. These mosquitoes can, therefore, be used to replace the existing *Ae. aegypti* population with a lower competence phenotype by releasing both females and males (known as an introgression or replacement intervention).

The World Mosquito Program (WMP) has partnered with governments and communities to deploy *Wolbachia* mosquitoes in 11 countries since 2011 [3]. A number of randomised and non-randomised field trials have been conducted [4–8] demonstrating successful establishment of *w*Mel in local *Ae. aegypti* populations and the efficacy of the intervention in controlling dengue and other *Aedes*-borne diseases. The Vector Control Advisory Group of the World Health Organization concluded in December 2020 that the evidence for *w*Mel introgression demonstrates public health value against dengue [9]. A past economic evaluation by Brady *et al.* [10] found that *Wolbachia* deployments (using a replacement intervention) were likely to be cost-effective for controlling dengue in urban areas of Indonesia.

Dengue has been estimated to cause a substantial health and economic burden in Vietnam [11]. In Vietnam, *w*Mel *Wolbachia* mosquito releases have been undertaken in several demonstration projects [12]. However, there remains a need to evaluate the health and economic benefits of such an intervention at a large scale, as well as its value for money.

In this paper, we evaluate the potential economic impact and cost-effectiveness of scaled *Wolbachia* deployments as a form of dengue control in Vietnam–targeted at the highest burden urban areas.

## Methods

### The selected settings and dengue incidence

Ten high dengue burden urban settings within Vietnam, including four metropolitan and six provincial cities, were identified as priority locations for potential future *Wolbachia* deployments (described in and S1 Appendix). These settings accounted for 38% of the dengue cases notified in Vietnam between 2016 and 2019, and approximately a quarter of the national population.

The assumed overall incidence of dengue in Vietnam was taken from the model-based estimates from the Global burden of disease (GBD) 2019 study (1,047,320 symptomatic dengue cases occurring in Vietnam) [13]. We based the relative sub-national distribution of this country-level incidence on the mapping estimates of Bhatt *et al.* [14].

The total number of symptomatic dengue cases were broken down into the following severity categories; sought no formal treatment, outpatient cases, hospitalized cases, and fatal cases (Table 1), based on the distribution reported previously from Indonesia [17] as no empirical data on this distribution were available from Vietnam.

### Health burden and economic burden of dengue

The DALY burden of non-fatal cases was estimated using the overall disability weights from Zeng *et al.* [18] (Table A in S1 Appendix). It was assumed that a self-managed case had the same disability weight as an outpatient case. The number of years of life lost per fatal case was estimated from the GBD 2019 study [13].

The investigated economic burden had two components. The first was the cost of illness associated with the dengue cases (this was stratified by direct medical costs, direct non-medical costs, and productivity costs). The second was the costs related to the government's standard dengue prevention and control activities. Further details are provided in the supporting information (Tables B and C in S1 Appendix).

### Costs of the *Wolbachia* deployments

The intervention was divided into the following phases: preparation phase, release phase, short-term monitoring phase—including any required re-release, and long-term monitoring phase (see S1 Appendix).

The costs for preparation, release and short-term monitoring phases were informed by the WMP's accounts from two project sites within Vietnam in 2022 (see S1 Appendix). The costs of the long-term monitoring phase were based on WMP's implementations in other countries (Table D in S1 Appendix). These long-term monitoring functions are assumed to be integrated into routine public health activities and require no or only very occasional procurement of additional resources.

Based on these assumptions the average total cost per person covered was assumed to be US$8.56 (discounted at 3% per year and expressed in 2020 US$ prices) (Table D in S1 Appendix). The total implementation costs were based on the population within the release area for each setting. It was assumed that the full cost of the intervention was incurred by the health care provider and that it was not influenced by the chosen perspective.

**Table 1. The settings included within this analysis.**

| Province | District | Reported Ministry of Health case numbers (average 2016–2019) | Projected annual number of dengue cases[2] | Intervention release area (km²) | Population within the intervention release area[3] | Total population based on the administrative district boundary[4] | Population density of the release area (population per km²) |
|---|---|---|---|---|---|---|---|
| Hồ Chí Minh | All | 13,714 | 161,582 | 607 | 8,881,693 | 8,993,082 | 14,630 |
| Hà Nội | All | 13,472 | 115,800 | 352 | 6,198,796 | 8,053,663 | 17,626 |
| Đà Nẵng | All | 4,455 | 12,329 | 114 | 1,052,136 | 1,134,310 | 9,265 |
| Cần Th | All | 1,004 | 19,366 | 119 | 983,972 | 1,235,171 | 8,269 |
| Bình Dng | Thuận An | 2,734 | 13,106 | 30 | 514,425 | 596,227 | 17,249 |
| Bình Dng | Dĩ An | 1,660 | 10,439 | 27 | 444,657 | 474,681 | 16,346 |
| Bình Dng | Th Dầu Một | 1,637 | 7,791 | 27 | 273,262 | 321,607 | 10,110 |
| Đồng Nai | Biên Hòa | 1,637 | 14,084 | 53 | 996,198 | 1,055,414 | 18,624 |
| Khánh Hòa | Nha Trang | 2,226 | 4,014 | 24 | 413,443 | 422,601 | 17,393 |
| Bà Rịa - Vũng Tàu | Vũng Tàu | 1,688 | 4,576 | 16 | 261,253 | 357,124 | 16,157 |

[1] Data on the number of dengue cases notified to national dengue surveillance system by district each year 2016–2019 was provided by the Department of Preventive Medicine of the Vietnam Ministry of Health.

[2] The assumed overall incidence of dengue for Vietnam was based on the GBD 2019 study (1,047,320 cases) [13]. The relative distribution of this overall burden to the different areas investigated was based on the mapping estimates of Bhatt et al. [14].

[3] Was estimated based on the proportion of the population living in the release area (according to the data from WorldPop [15]) and the total population of the administrative district boundary according to the 2019 census data [16].

[4] Based on data from the 2019 census [16].

### Effectiveness of the *Wolbachia* deployments

Based on data from a cluster randomized trial and quasi-experimental studies in Yogyakarta, the effectiveness of *Wolbachia* deployments in reducing the incidence of symptomatic cases was assumed to be 75% [5,7]. It was further assumed that there is a lag of six months after the completion of releases before *Wolbachia* reaches a stably high prevalence in the local *Ae. aegypti* population, and for the full intervention effect to accrue.

We assume the *Wolbachia* intervention's effect is area-wide and benefits the whole population within the administrative district boundary (see S1 Appendix) and that the benefit of the intervention was equally experienced among the targeted populations.

### The economic evaluation

To evaluate the intervention a scenario with scaled *Wolbachia* deployments in the selected settings was compared incrementally to an alternative scenario of continuing existing dengue control measures (i.e. the comparator was the status quo). It was assumed that in the absence of the intervention the baseline number of dengue cases occurring would increase annually based on the population growth rate of 1.14% (the average for Vietnam) [19]. The calculations compared the number of dengue cases (and their associated health and economic burden) that would be occurring under the status quo comparator compared to the number projected to be occurring in the presence of *Wolbachia* deployments. This was calculated with a static based approach, as the indirect effects of the intervention were accounted for within the population-level effectiveness metric.

For the base case results, we assumed that the intervention would have its full impact for 20 years once the intervention becomes effective—six months after the completion of releases (therefore the total time horizon of the analysis was 22 years, to account for the 18-month preparation and release phases). We assumed the intervention was implemented in each setting independently, but for simplicity no specific assumptions were made on the sequence or timeline of releases across the ten settings.

All costs and benefits are given in 2020 US$ prices and were discounted at 3% per year (in line with WHO-CHOICE recommendations) [20]. The different outputs and perspectives considered are summarised in Box 1. The cost-effectiveness ratios were compared to a cost-effectiveness threshold of 0.5 times Vietnam's GDP per capita (i.e. <US$1,760 per DALY averted) [21,22]. A CHEERS checklist [23] is provided in the S1 Checklist file. Univariate sensitivity analysis on several parameters summarised in Table 2.

## Results

### Baseline burden

It was projected that across the ten cities, an average of 363,086 symptomatic dengue cases occurred each year at baseline of which 42 (0.01%) would be fatal (S1 Table). Stratified by severity, we estimated 232,738 (64%) of the cases sought no formal treatment, 84,599 (23%) cases only sought treatment at outpatient facilities and 45,749 (13%) cases would be hospitalized. The corresponding baseline DALY burden was 13,674 (S1 Table).

Our projected number of hospitalized cases is generally comparable with the number reported to the Ministry of Health (45,749 vs 36,754 cases per annum) (S2 Table). Although this is promising as an average, there was variation across the different settings in the consistency between model-based and reported dengue case numbers (S2 Table).

The total corresponding baseline cost of illness associated with the dengue cases totalled US$23.69 million per year (S1 Table). Half of this was from productivity costs. In addition to this cost of illness, an estimated US$809,105 was spent each year on the government's current dengue prevention and control activities–giving a total baseline economic burden of US$24.50 million per year (S1 Table). 19% of the economic burden was incurred by the health system and 81% by the patients.

### Cost of the *Wolbachia* deployments

The discounted total cost of *Wolbachia* implementation across all ten settings was projected to be US$171.3 million. The cost for each of the settings is shown in S3 Table. The majority of the cost was incurred during the preparation and release phases, occurring within the first 2 years of the intervention (Fig 1 and Table D in S1 Appendix).

### Impact of the *Wolbachia* deployments

Assuming that the intervention has 20 years of impact once *Wolbachia* is established in the local mosquito population, we estimated that the intervention would avert 6.2 million dengue cases including over 784,000 hospitalisations, result in 153,285 DALYs being averted (i.e. healthy life-years gained), and generate US$299 million in economic benefits (S3 Table). The majority of the benefits were experienced by the patients (US$245 million) but the benefit to the health system was also notable (US $54 million). Setting-specific benefits are shown in S3 Table.

## Box 1. Output of the economic evaluation

### Cost-utility analysis

- Incremental cost-effectiveness ratios: these ratios were considered incrementally from the comparator. The health benefits were measured with the DALYs averted metric. Within these, the relevant cost savings compared to the comparator scenario were included and deducted from the cost of the intervention (such as averted costs associated with prevented hospitalised cases). Which cost offsets/savings are included depends on the perspective of the analysis:

  - Healthcare provider perspective: only savings related to averted direct medical costs that are incurred by the healthcare providers and averted costs related to the government's current dengue prevention and control activities costs are considered.

  - Health sector perspective: only savings related to averted direct medical costs (from both the patients and the healthcare providers) and averted costs related to the government's current dengue prevention and control activities costs are considered.

  - Societal perspective: in addition to the savings in averted direct medical costs, the savings related to the patients' prevented direct non-medical costs (such as transport to the hospital/clinic) and the estimated monetary value of the prevented productivity losses that would have been associated with a dengue case. Due to ongoing debates in this area [27,28], the results from the societal perspective were shown both including and excluding the productivity gains related to prevented premature mortality.

- We also report the estimated gross cost-effectiveness ratios where the investment cost of the intervention is simply divided by the number of DALYs averted (and no cost savings/offsets were considered). These are summarised to allow comparison to other studies reporting this metric.

### Cost-benefit analysis

- Benefit-cost ratios: the projected economic benefit of the intervention is divided by the investment cost of the intervention. When this ratio is above one it indicates that the economic benefits generated outweigh the costs of the intervention. These were considered from the societal perspective. The economic benefits were based on the averted cost of illness from the prevented dengue cases (the averted direct medical and non-medical costs and the averted productivity costs) and averted costs related to the government's current dengue prevention and control activities costs. The number of DALYs averted were not monetised within these calculations.

**Table 2. Parameter table.**

| Parameter | Base case | Range investigated in the sensitivity analysis |
|---|---|---|
| Epidemiological assumptions | | |
| Baseline burden of dengue (the annual number of symptomatic dengue cases occurring across Vietnam) | 1,038,968 [13] | 812,231–1,333,875 [13] |
| Annual growth in case numbers | 1.14% [20] | 0% up to 5% |
| Death rate per 100,000 symptomatic dengue episodes | 11.65 [13] | - |
| Breakdown of cases | | |
| Would not seek formal treatment | 64.1% [17] | 7% [24] |
| Would seek formal treatment as an outpatient | 23.3% [17] | 76% [24] |
| Would seek formal treatment and be hospitalized | 12.6% [17] | 17% [24] |
| DALY | | |
| DALYs lost per non-fatal case that sought no formal treatment | 0.0307 [18] | 0.0107 [18] |
| DALYs lost per non-fatal outpatient case | 0.0307 [18] | 0.0107 [18] |
| DALYs lost per non-fatal hospitalized case | 0.0351 [18] | 0.0152 [18] |
| Average years of life lost per fatal case (undiscounted) | 55 [13] | - |
| Cost of illness per case | | |
| Cost per case that sought no formal treatment | US$17.32 [24] | US$13.01–22.88 [25] |
| Cost per outpatient case | US$69.03 [25] | US$54.56–90.18 [25] |
| Cost per hospitalized case | US$223.79 [25] | US$171.46–283.91 [25] |
| Cost per fatal case | US$84,901.79[1] | - |
| Proportion of the direct medical costs of hospitalized and outpatient cases would be incurred by the healthcare provider | 55.3% [25] | - |
| Government's current dengue prevention and control activities | | |
| Costs of the government's current dengue prevention and control activities. | See Table C in S1 Appendix | |
| Reduction in the current dengue prevention and control activities once the *Wolbachia* deployments have become effective | 75% | |
| Cost of the intervention | | |
| Cost per person covered (discounted) | US$8.56 per person covered | US$3–12 per person covered |
| Effectiveness of the intervention and discount rates | | |
| Duration of the impact of the *Wolbachia* deployments[4] | At least 20 years [26] | 10–25 years |
| Effectiveness of the *Wolbachia* deployments in terms of reducing the number of cases for the targeted setting | 75% [7] | 65–85% [5] |
| Discount rates (for costs and effects) | 3% [20] | 0% for the health benefits [20] 6% for the costs [20] |

[1] Estimated using the human capital approach (see S1 Appendix).

[2] See Table B in S1 Appendix for the data stratified by cost type.

[3] See Table D in S1 Appendix for the data stratified by the phase of the intervention.

[4] Considered from once the intervention becomes effective—6 months after the completion of releases.

## The cost-effectiveness of the *Wolbachia* deployments

The projected cost per DALY averted was notably influenced by the perspective of the analysis (Table 3). That said, regardless of the perspective used, all of the overall cost-effectiveness ratios (i.e. when aggregated across all of the settings) were under the cost-effectiveness

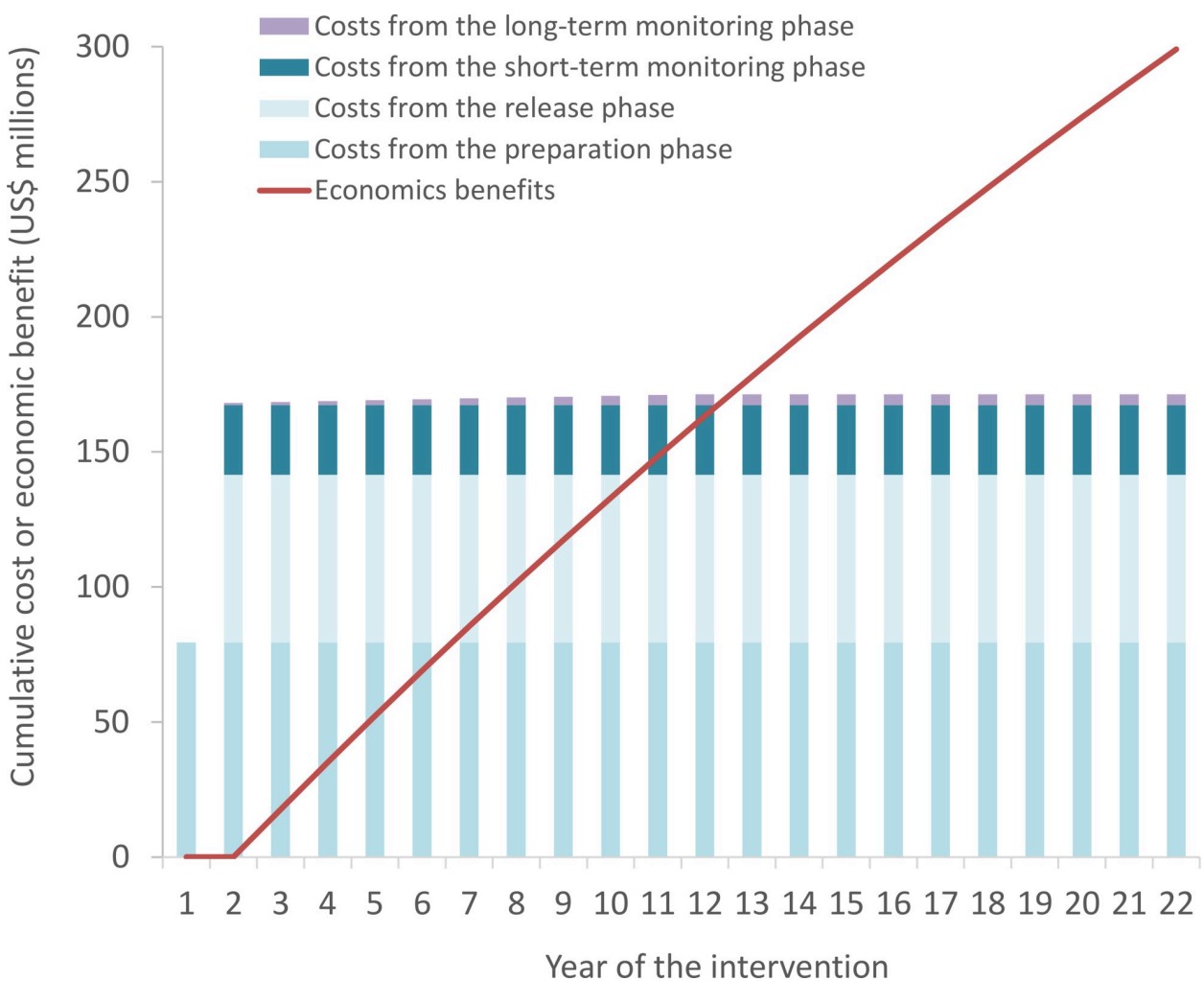

**Fig 1. Distribution of cumulative costs and benefits.** All costs and benefits are in present-day value 2020 US$ discounted at 3% per year.

threshold (<US$1,760 per DALY averted) (S4 Table). From the health sector perspective, the cost per DALY averted was US$420. From a societal perspective, the cost-effectiveness ratio was negative (i.e. the economic benefits outweighed the cost)–even when excluding the productivity gains related to prevented excess mortality. The societal benefit-cost ratio was 1.75, meaning that the valued economic benefits are larger than the cost of the intervention over a 22-year horizon (Table 3 and Fig 1).

Note that there was variation in the setting-specific cost-effectiveness ratios (S4 Table). The intervention was the least cost-effective in the settings with lowest dengue incidence and vice versa.

## Sensitivity analysis

We performed univariate sensitivity analysis on several parameters (Table 2). The results of this are shown in Fig 2 and S5–S7 Tables. The results were most sensitive to the following parameters:

Time horizon and duration of the effectiveness: If the intervention was assumed to only generate 10 years of benefits, the projected long-term impact decreased. Under this scenario,

**Table 3. Impact of the *Wolbachia* deployments stratified by cost type.**

| Economic benefits over the full time horizon | Total benefit across the settings (US$ 2020 prices) |
|---|---|
| Costs related to cases that sought no formal treatment averted | 49,369,662 |
| Costs related to outpatient cases averted | 71,530,105 |
| Costs related to hospitalized cases averted | 125,402,979 |
| Costs related to fatal cases averted | 43,983,937 |
| Direct medical costs averted | 98,236,369 |
| Direct non-medical costs averted | 42,202,069 |
| Productivity costs averted | 149,848,245 |
| Health system costs averted | 53,989,360 |
| Patients' costs averted | 245,062,445 |
| Costs averted related to the government's current dengue prevention and control activities | 8,765,122 |
| Total cost of illness averted | 290,286,683 |
| Total economic burden averted | 299,051,805 |
| **Cost-effectiveness/benefit-cost ratios** | **Cost per DALY averted (2020 US$ prices)** |
| ICER—health care provider perspective | 708.21 |
| ICER—health sector perspective | 419.56 |
| ICER—societal perspective | "Cost saving" (-776.16)[1] |
| ICER—societal perspective (excluding the productivity gains related to prevented excess mortality) | "Cost saving" (-546.40)[1] |
| Societal benefit-cost ratio | 1.75 |

The gross cost-effectiveness ratio is presented in S5–S7 Tables. ICER: Incremental cost-effectiveness ratio.

[1] Negative ratios ("Cost savings") in the case indicate that the economic benefits of the health intervention relative to the comparator outweighed the cost of the intervention. Note that these "Cost savings" include non-fiscal costs.

the ICERs in the majority of individual settings as well as the overall cost-effectiveness ratios (i.e. when aggregated across all of the settings) remained under the cost-effectiveness threshold (S6 Table). However, under this assumption for some of the individual settings, the ICERs exceeded the cost-effectiveness threshold (S6 Table).

The DALY disability weights and the inclusion of persistent symptoms: Using disability weights that only accounted for acute symptoms of dengue notably increased the cost-effectiveness ratio from the health sector perspective. However, from a societal perspective, the ratios remained negative for the majority of the settings.

Annual growth in case numbers: For the base case results we assumed a modest 1.14% increase in the baseline case burden of dengue (based on the average population growth rate for Vietnam). However, under a scenario of a higher increased per capita dengue incidence over time, this notably increases the projected value for money of the intervention

Cost per person covered: Intuitively the projected average cost-effectiveness and cost-benefit of the *Wolbachia* deployments were sensitive to assumptions regarding the cost per person covered (S7 Table). This highlights that opportunities for further reductions in the cost of the intervention would make it even more cost-effective.

## Discussion

Dengue is a major public health problem worldwide and its burden is expected to increase even further due to climate change and urbanization [1]. It is therefore vital that new dengue

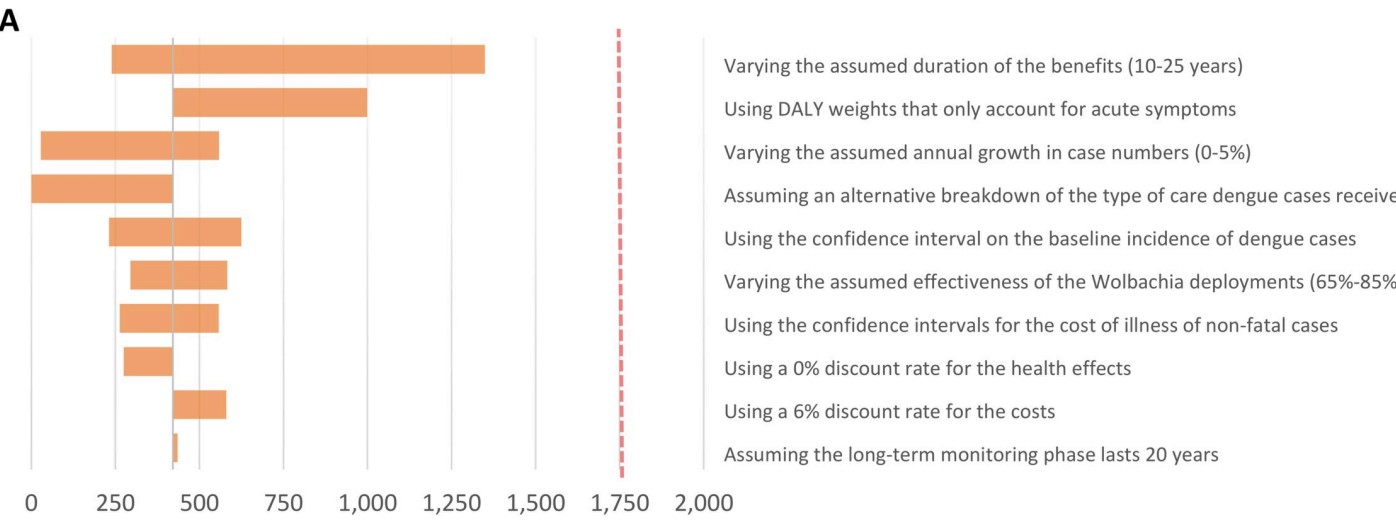

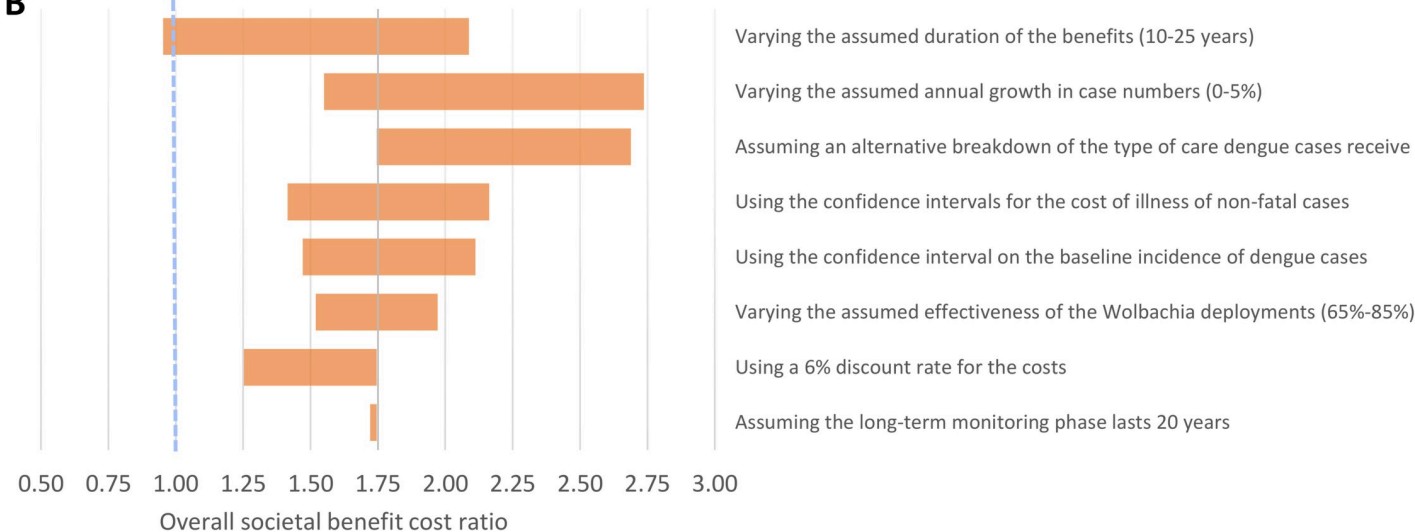

**Fig 2. Tornado plot illustrating the impact of the sensitivity analysis on overall cost-effectiveness from the health sector perspective and overall societal benefit cost ratio Red dashed line indicates the cost-effectiveness threshold of 0.5 times Vietnam's GDP per capita (i.e. <US$1,760 per DALY averted).** The blue dashed line indicates a societal benefit cost ratio of one. The ranges investigated are provided in Table 2.

interventions are developed and evaluated [29,30], such as the introduction of *Wolbachia* into *Ae. aegypti* mosquito populations. Health economic analyses play an important role here in evaluating the costs and benefits of new interventions to investigate their value for money and inform health policy. In this analysis, we investigated the cost-effectiveness of releasing *Wolbachia* mosquitoes throughout ten high burden cities in Vietnam as a form of dengue control. These settings account for approximately 40% of the country's reported national dengue burden and we estimated that the *Wolbachia* intervention could avert 6.2 million dengue cases over the next two decades, generating substantial health and economic benefits. Overall this was found to be a cost-effective intervention and these results support previous findings that *Wolbachia* deployments can be a cost-effective intervention when targeted to high burden urban settings [10].

We estimated that the total cost of implementing *Wolbachia* in these ten cities would be US $171.3 million. Importantly, the majority of this cost was incurred during the preparation and release phases occurring within the first two years of the intervention. In contrast, the benefits accrued over 20 years (Fig 1). Our cost projections for the *Wolbachia* deployments were driven by the projected cost per person covered from two WMP demonstration projects in the south of Vietnam and were intended to be conservative. However, it is important to note that there is a degree of uncertainty regarding what the long-term costs of *Wolbachia* deployments would be in practice if it was implemented programmatically at this scale, and such cost estimates are sensitive to assumptions regarding the level of technical and implementation support that would be required from WMP global staff for subsequent releases. There is potential for the costs of *Wolbachia* deployments to be reduced over time, through advances in mass mosquito production, economies of scale, and alternative implementation models.

The estimated cost-effectiveness ratios were highly dependent on the perspective of the analysis and what savings were considered (Box 1). This needs to be considered when comparing the results to other studies. Averaged across all ten settings, the cost per DALY averted was US$420 from the health sector perspective. In contrast, from a societal perspective (which accounts also for averted productivity losses), the cost-effectiveness ratios were negative for the majority of the settings, meaning that the projected economic benefits generated by the intervention over 20 years outweighed its cost. Note that there are issues around potentially double counting benefits when including productivity gains within cost-effectiveness ratios [27,28]. Due to this we also calculated the societal perspective excluding the productivity gains related to prevented excess mortality and the results were still favourable.

In terms of policy recommendations based on international standards for defining cost-effectiveness, the values were below the conservative 0.5 times GDP per capita threshold [21,22] in all of the settings when using the health sector perspective–and mostly negative (i.e. the economic benefits outweigh the cost) from a societal perspective (S4 Table). In addition, the overall average cost-effectiveness ratio using the health sector perspective was below the threshold of US$500 per DALY averted set by the third edition of the disease control priorities project to identify priority interventions for consideration in lower-middle-income countries [31]. The results were also favorable when compared to the cost-effectiveness of a diverse range of health interventions in Vietnam within the Global Health Cost Effectiveness Analysis (GH CEA) Registry [32]. This indicates that the intervention offers good value for money in this setting. The results were most sensitive to assumptions related to the duration of the impact of the intervention, the exclusion of persistent symptoms within the disability weights, the assumed growth in case numbers and the *Wolbachia* deployment costs.

Due to the sensitivity of these results to the assumed incidence of dengue and the cost of illness associated with dengue cases, the conclusions cannot be directly generalised to other country settings and further evaluations are needed. That said, the overall conclusions were similar to an economic evaluation of *Wolbachia* deployments in Indonesia [10], with the key difference being that the Indonesian study only considered benefits up to 10 years post completion of releases for their primary results whereas we assumed 20 years. This longer duration of benefit is supported by a modelling study [26] and by field evidence a decade after the first *Wolbachia* release sites in northern Australia showing stability of *Wolbachia* in the *Ae. aegypti* population and maintenance of its virus-blocking properties [33]. This assumption was varied in our sensitivity analysis, and even with a more conservative assumption of 10 years of benefits we predict that, overall, the *Wolbachia* intervention would remain cost-effective in Vietnam, though less so than predicted in the Indonesian setting. This could be because the cost per dengue case is higher in Indonesia, which therefore means averted cases generate larger savings to offset the cost of the intervention.

In addition, Box 2 highlights some of broader benefits of the intervention not accounted for within this analysis.

Although these findings are positive it is important to consider that there will be no one solution to controlling dengue and it remains vital to consider/evaluate other interventions (such as new vaccines as they become available [38]).

## Key assumptions and limitations

Among several important assumptions and limitations of this analysis (also discussed further in S1 Appendix) is the uncertainty regarding the baseline burden of dengue in the settings considered. The use of model-projected estimates of dengue burden is justified by the known large under-ascertainment of dengue in empirical disease surveillance data [39], but if these model-projected estimates are overestimated or overgeneralised, this would consequently overestimate the impact and cost-effectiveness of the *Wolbachia* deployments. That said, the GBD 2019 study estimate of 1,047,320 average annual national dengue cases used is lower than some of the other estimates for Vietnam–with past estimates of over 2 million cases per year [11]. A further limitation was that the breakdown of non-fatal cases was based on data from Indonesia and was not specific to Vietnam [17].

The same estimated cost per person covered was used for all of the project sites. However, in reality, this would vary due to economies of scale and the population density of each setting (see S1 Appendix). In addition, for simplicity we assumed that the intervention was implemented independently in each setting. However, Brady *et al.* [10] found that a "sequenced" delivery scenario, where the releases are spread over a period of time with certain centralised resources reutilised across different locations could be cheaper. Costing *Wolbachia* deployments under different scenarios of scaled production and implementation is an important area for further investigation.

Several factors could theoretically lead to a lower long-term effectiveness of *Wolbachia* deployments [10]. These include reinvasion by *Wolbachia* uninfected mosquitoes, evolution of viral resistance, temperature effects on viral blocking efficacy and inheritability, and selection of more virulent dengue virus strains. In addition, the successful dispersal of *Wolbachia* mosquitoes can be heterogeneous and influenced by local environmental factors [10, 12]. This can lead to pockets of low *Wolbachia* frequency, reducing the impact of the intervention. To account for this possibility of heterogeneous *Wolbachia* introgression in some locations, we used an estimate of effectiveness consistent with that measured in a quasi-experimental study [7] and cluster randomized trial in Yogyakarta [5], but which is lower than some of the model-based estimates of effectiveness [17] and likely conservative for a scenario of city-wide deployment where the diluting effects of human movement outside the *Wolbachia* release area are minimized. The sustainability of the long-term (>10 years) effectiveness of *Wolbachia* deployments is an area that requires further investigation.

Finally, there is a notable variation in the disability weights used to calculate the DALYs lost due to dengue [11,40]. For consistency with the other economic evaluation of *Wolbachia* deployments we used weights estimated by Zeng *et al.* [18]. These are higher than the weights used for dengue by the GBD. A notable source of uncertainty surrounding these disability weights is the level and duration of any persistent symptoms of dengue [40–42]. When using the disability weights that only accounted for the acute symptoms it notably increased the cost-effectiveness ratios from the health sector perspective to US$999 per DALY averted. However, the overall ratio remained under the cost-effectiveness threshold and when using the societal perspective, the ratios for the majority of the settings remained negative.

## Box 2. Broader benefits of the intervention

Our findings highlight the projected benefits to the health system of implementing *Wolbachia* for dengue control in Vietnam, through averted outpatient and inpatient dengue cases. There would also be notable social-economic benefits–including millions of productive days gained by both the patients and their caregivers. Because *Wolbachia* is deployed at a community level, all sub-populations benefit from the intervention, including socio-economically disadvantaged groups who are disproportionately affected by the economic burden of dengue.

As well as the investigated benefits associated with controlling dengue related to averted cost of illnesses there are other potential economic benefits. For example, dengue can lead to lost revenue from tourism [34]. Controlling dengue with *Wolbachia* could therefore lead to economic benefits from increased tourism in a setting like Vietnam [11] and potentially offset some of the costs of the intervention.

Dengue outbreaks are typically seasonal, with a notable proportion of the cases occurring over a period of several months. This means that hospitals, and particularly intensive care wards, may become congested during dengue outbreaks. It is possible that this could have negative consequences on care for patients with other conditions due to deterioration of overall service quality.

In this economic evaluation, we have only considered the dengue burden avertable by *Wolbachia* deployments. However, *Wolbachia* mosquitoes are also refractory to Zika virus, chikungunya virus and, yellow fever virus [35,36]. This means that the overall public health impact of *Wolbachia* deployments will be larger, but the sporadic nature of these other epidemic arboviruses makes it challenging to quantify these projected benefits.

It is also important to note that climate change is likely to expand the geographical distribution and transmission intensity of several vector-borne human infectious diseases–including dengue [37]. This, as well as population growth and other factors, could potentially increase the incidence of dengue and other epidemic arboviruses, increasing the burden that would be averted by *Wolbachia* deployments in Vietnam.

## Conclusions

Overall, we found targeted deployments of *Wolbachia* in high dengue burden cities would be a cost-effective intervention for dengue control in Vietnam, generating considerable health and economic benefits from both a health sector and societal perspective. Our primary results are based on an assumption that the long-term effectiveness of *Wolbachia* releases is sustained for 20 years, but we predict that *Wolbachia* deployments in Vietnam would overall remain classed as cost-effective in the majority of the settings even considering a more conservative time horizon of 10 years of benefits. Overall, this work highlights the value of investment in the scaled implementation of *Wolbachia* deployments as an effective and cost-effective tool for dengue control in Vietnam, and more generally for addressing the global challenge of dengue control.

## Supporting information

**S1 Appendix. Supporting information. In this supporting information, we provide additional methodological information.**
(DOCX)

**S1 Checklist. A CHEERS 2022 checklist.**
(DOCX)

**S1 Table. Baseline burden of dengue across the study settings.**
(DOCX)

**S2 Table. Comparison of the projected number of hospitalized cases to Ministry of Health data.**
(DOCX)

**S3 Table. Base case projected total cost and impact of the Wolbachia deployments (2020 US$ prices).**
(DOCX)

**S4 Table. Base case setting specific cost-effectiveness ratios (2020 US$ prices.**
(DOCX)

**S5 Table. Results of the sensitivity analysis (2020 US$ prices).**
(DOCX)

**S6 Table. Setting specific cost-effectiveness ratios–when assuming only 10 years of benefits (2020 US$ prices).**
(DOCX)

**S7 Table. The cost-effectiveness and cost benefit of the Wolbachia deployments for different intervention costs.**
(DOCX)

## Author Contributions

**Conceptualization:** Hugo C. Turner, Cameron P. Simmons, Katherine L. Anders.

**Data curation:** Duong Le Quyen, Reynold Dias, Phan Thi Huong.

**Formal analysis:** Hugo C. Turner.

**Writing – original draft:** Hugo C. Turner.

**Writing – review & editing:** Duong Le Quyen, Reynold Dias, Phan Thi Huong, Cameron P. Simmons, Katherine L. Anders.

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
