## [Decision Letter · Decision Letter 0]

6 May 2023

Dear Dr. Turner,

We are pleased to inform you that your manuscript 'An economic evaluation of Wolbachia deployments for dengue control in Vietnam' has been provisionally accepted for publication in PLOS Neglected Tropical Diseases.

Best regards,

Rebecca C. Christofferson

Academic Editor

Nigel Beebe

Section Editor

Reviewer's Responses to Questions

**Key Review Criteria Required for Acceptance?**

**Methods**

-Are the objectives of the study clearly articulated with a clear testable hypothesis stated?

-Is the study design appropriate to address the stated objectives?

-Is the population clearly described and appropriate for the hypothesis being tested?

-Is the sample size sufficient to ensure adequate power to address the hypothesis being tested?

-Were correct statistical analysis used to support conclusions?

-Are there concerns about ethical or regulatory requirements being met?

Reviewer #1: The study design, statistical analysis, and other parameters used satisfy all necessary requirement.

Reviewer #2: (No Response)

**Results**

-Does the analysis presented match the analysis plan?

-Are the results clearly and completely presented?

-Are the figures (Tables, Images) of sufficient quality for clarity?

Reviewer #1: the data presentediun the different forms match the analysis plan

Reviewer #2: (No Response)

**Conclusions**

-Are the conclusions supported by the data presented?

-Are the limitations of analysis clearly described?

-Do the authors discuss how these data can be helpful to advance our understanding of the topic under study?

-Is public health relevance addressed?

Reviewer #1: the team indicated their limitations, outlined recommendations and presented enough data to make their conclusions

Reviewer #2: (No Response)

**Editorial and Data Presentation Modifications?**

Reviewer #1: (No Response)

Reviewer #2: Spelling error in the author summary.

**Summary and General Comments**

Reviewer #1: The authors set out to evaluate the economic benefit of deploying Wolbachia in central Vietnam and as of reducing Dengue fever occurrence.

The claims are novel and can be used by policy makers in areas where Dengue fever is endemic. Also, this can be replicated for other diseases that are vectored by the Aedes spp.

the authors treated the literature fairly and have enough data to support their claims. methods are thoroughly explained and supported with adequate reference.

What stands out about the paper is their use of predictive models in Wolbachia deployment. Evidence of substantial results indicative by the supporting. Paper is properly organized as well.

Reviewer #2: Well written.

PLOS authors have the option to publish the peer review history of their article (what does this mean?). If published, this will include your full peer review and any attached files.

Reviewer #1: No

Reviewer #2: No

---

## [Editor Report · Acceptance letter]

25 May 2023

Dear Dr. Turner,

We are delighted to inform you that your manuscript, "An economic evaluation of Wolbachia deployments for dengue control in Vietnam," has been formally accepted for publication in PLOS Neglected Tropical Diseases.

Best regards,

Shaden Kamhawi

co-Editor-in-Chief

Paul Brindley

co-Editor-in-Chief
